# ARKVALE: Efficient Generative LLM Inference with Recallable Key-Value Eviction

**Renze Chen**
Peking University
crz@pku.edu.cn

**Zhuofeng Wang**
Peking University
2200012827@stu.pku.edu.cn

**Beiquan Cao**
Peking University
2200012988@stu.pku.edu.cn

**Tong Wu**
Peking University
2200013212@stu.pku.edu.cn

**Size Zheng**
Peking University
zhengsz@pku.edu.cn

**Xiuhong Li**
Peking University
lixiuhong@pku.edu.cn

**Xuechao Wei**
Peking University
xuechao.wei@pku.edu.cn

**Shengen Yan**
Infinigence-AI
yanshengen@gmail.com

**Meng Li**
Peking University
meng.li@pku.edu.cn

**Yun Liang**[*]
Peking University
ericlyun@pku.edu.cn

## Abstract

Large Language Models (LLMs) are widely used in today's tasks of natural language processing. To support applications like multi-turn chats, document understanding, and content generation, models with long context lengths are growing in importance. However, managing long contexts brings substantial challenges due to the expansion of key-value cache (KV cache). Longer KV cache requires larger memory, limiting the batch-size and thus decreasing throughput. Also, computing attention over long KV cache incurs more memory access, hurting the end-to-end latency. Prior works find that it is sufficient to use only the recent and high-impact tokens for attention computation, allowing the eviction of less vital tokens to reduce memory footprint. Nonetheless, we observe a dynamic shift in token importance across different decoding steps. Tokens initially evicted might regain importance after certain decoding steps. To address this, we propose ARKVALE, a page-based KV cache manager that can recognize and recall important tokens evicted before. We asynchronously copy the filled page into external memory (e.g., CPU memory) as backup and summarize/compress it into a much smaller digest by constructing the bounding-volume of the keys in the KV-page. Before attention computation, we measure all pages' importance based on their digests, recall the important ones, evict the unimportant ones, and select the top-ranked pages for attention computation. Experiment results show that ARKVALE performs well on various long context tasks with negligible accuracy loss under 2k∼4k cache budget and can improve decoding latency up to $2.2\times$ ($1.7\times$ in average) and batching throughput up to $4.6\times$ ($3.5\times$ in average). Our code is now available at https://github.com/pku-liang/ArkVale.

---

[*]Corresponding author.

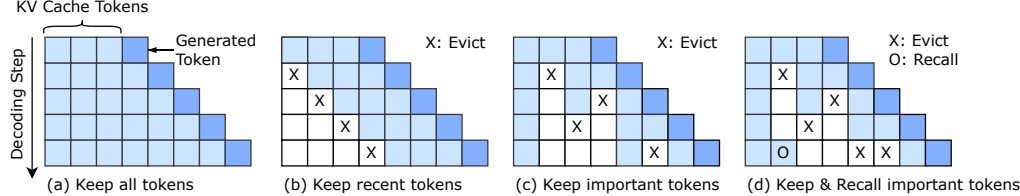

Figure 1: Comparison of different KV cache eviction works.

# 1   Introduction

Large Language Models (LLMs) are rapidly gaining universal presence, underpinning a myriad of natural language processing applications, including dialogue systems [2, 52, 16], document summarization [62, 24], code completion [13, 46], and question answering [31]. The context length supported by large models is also growing progressively to support more applications like multi-turn chat [2, 52, 16] and text summarization [28, 68, 22]. This shift has seen an expansion from an initial range of up to 16k, advancing steadily towards lengths of 32k [1], 128k [43], and even 2048k [21].

However, handling long contexts poses significant challenges. The key-value cache (KV cache) of LLMs scales with both the batch size and the length of historical context. For each token generated by an LLM, the query must attend to the entire set of context key-values. On one hand, longer KV caches occupy more memory spaces, limiting batch-size as well as throughput. On the other hand, computing attention with long KV cache incurs more memory accesses, hurting the inference latency.

Previous works [58, 64, 37, 23, 40, 6] reveal sparsity in KV cache, which means that only a subset of tokens in the KV cache significantly influence the accuracy of attention computation. They propose two characteristics of token importance: locality and persistence. Locality implies that recent tokens tend to be important for current attention, while persistence suggests that tokens previously deemed important are likely to retain their importance over time. They devise algorithms to retain recent tokens [58, 37, 64] (Figure 1 (b)) or vital tokens [58, 37, 64, 40, 23, 6] (Figure 1 (b)) while evicting older or less crucial ones to reduce memory usage of KV cache and memory accesses of attention.

In this work, we delve deeper into the dynamism of token importance, observing that tokens previously deemed unimportant may regain importance over time and vice versa. Previous works ignore this and may risk permanently discarding the tokens that are vital later on, inadvertently decreasing model accuracy. To address these challenges, we propose ARKVALE, inspired by vLLM [36] to organize tokens into pages for fine-grained management of KV cache. Upon filling a KV-page, we asynchronously copy it to CPU memory as a backup, and summarize/compress it into a much smaller digest by constructing a bounding-volume of the keys of the KV-page. Before each attention computation, we dynamically estimate the importance of each page based on their digests and current query, and selectively recall & evict some pages based on their importance scores (Figure 1 (d)).

Our contribution can be summarized as follows:

- We characterize the dynamism of the importance of tokens in the KV cache of LLM, and find that some unimportant tokens may regain importance over time.

- We propose ARKVALE, a KV cache manager which organize tokens into pages and dynamically evict & recall them based on their importance.

- We propose a method based on bounding-volume to summarize the pages and estimate the importance of them with a given query.

We evaluate ARKVALE against many state-of-the-art KV cache eviction works on various long-context benchmarks. Experimental results show ARKVALE performs well on all the long context tasks with negligible accuracy loss under a cache budget of 2k~4k and speedup model decoding latency up to $2.2\times$ ($1.7\times$ in average) and batching throughput up to $4.6\times$ ($3.5\times$ in average) in long context scenarios. Our code is now available at `https://github.com/pku-liang/ArkVale`.

## 2 Related Work

Many works have been proposed to extend context window of pre-trained models. One prevalent method is the integration of Rotary Position Embeddings (RoPE) [49]. Fine-tuning RoPE's scaling enables the Llama-2 model [55], initially handling 4k tokens, to be expanded to support 32k tokens in LongChat [1] and 128k tokens in Yarn-Llama-2 [43]. Recently, LongRoPE [21] pushes this boundary even further to accommodate up to 2048k tokens. However, as models become more capable of dealing with long context, they also face new challenges in terms of memory usage and inference efficiency. ARKVALE is designed to address such issues.

Some training-aware methods have been proposed to handle these problems. Multi-Query Attention (MQA) [47] and Group-Query Attention (GQA) [7] aim to train LLMs with fewer attention heads in KV cache. Works like RWKV [42], RetNet [50], and Mamba [25] propose Linear Attention/RNN to limit KV cache size. Sparse attention architectures [61, 10, 56, 17, 45, 33, 53] design special sparse pattern of attention during training to control the KV cache size during inference. But these works require much training effort for pre-training or fine-tuning while ARKVALE is a training-free method.

There are also some post-training methods, which evict tokens in KV cache to reduce memory usage and accelerate attention computation. StreamingLLM [58] and LM-Infinite [26] only keep initial tokens and recent tokens to maintain a fixed-size KV cache. Methods like H2O [64], Scissorhands [37] and TOVA [40] keep important tokens in KV cache based on the historical or current attention scores. FastGen [23] categorizes tokens and employs a more nuanced approach for choosing which KV cache tokens to preserve. Keyformer [6] improves the eviction score function by leveraging the Gumbel distribution. Q-Hitter [63] combines KV-cache quantization with KV-cache eviction. These methods rely on historical data to dictate cache eviction and may risk discarding the tokens that are important in the future. Works like IceFormer [38], SparQ [5], ALISA [65], and Quest [51] use post-training sparse attention to alleviate such issue but they require all KV cache residing in memory and thus cannot control KV cache size to optimize memory usage. Many scheduling-based methods [19, 18, 60, 15, 66, 14, 36, 48, 32, 27, 67] are also employed to optimize memory efficiency for LLMs or other DNNs, aiming to enhance memory usage and execution latency through proper computation partitioning and scheduling.

## 3 Background

### 3.1 Attention Computation and Generative Inference of LLM

The attention computation in a typical LLM involves mixing token-level information of queries $\mathbf{Q} \in \mathbb{R}^{s_q \times d}$ and keys $\mathbf{K} \in \mathbb{R}^{s_{kv} \times d}$ with values $\mathbf{V} \in \mathbb{R}^{s_{kv} \times d}$, where $d$ is the hidden-dimension and $s_q$ ($s_{kv}$) means the number of query (key/value) tokens. The attention computation can be formulated as: $\mathbf{S} := (\mathbf{Q} \cdot \mathbf{K}^\top / \sqrt{d}) \in \mathbb{R}^{s_q \times s_{kv}}$, $\mathbf{P} := \texttt{softmax}(\mathbf{S}) \in \mathbb{R}^{s_q \times s_{kv}}$, $\mathbf{O} := (\mathbf{P} \cdot \mathbf{V}) \in \mathbb{R}^{s_q \times d}$, where $\mathbf{P}$ is often called "attention map/scores" and $\mathbf{P}_{i,j}$ reflects the importance of key/value token $j$ to query token $i$. Then, the generative inference process of LLM mainly consists of two stages: the prefill/prompt stage and the decoding/generation stage. The prefill stage takes a prompt sequence with length $s_{\text{in}}$ as input and caches the keys & values computed by each layer of the LLM. The decoding stage uses and updates the KV cache to generate tokens step-by-step, where the current token depends on past tokens' keys & values stored in the KV cache. For the attention in prefill stage, $s_{\text{in}} = s_q = s_{kv}$, while $s_q = 1$ and $s_{kv} = \#\text{past-tokens} + 1$ in each decoding step.

### 3.2 Impact of Long Context

For a decoding step with batch-size as $b$, number of transformer layers as $l$, history sequence length as $s$, hidden-dimension as $h$, and data type as `float16`, the KV cache requires $4bsh$ bytes memory accesses for each attention computation and occupies $4blsh$ bytes of memory in total. The latency & memory breakdown of a decoding step of model LongChat-7b-v1.5-32k [1] ($l = 32, h = 4096$) with $b = 8$ and $s = 2^9, 2^{10}, 2^{11}, 2^{12}, 2^{13}, 2^{14}, 2^{15}$ is shown in Figure 2. As the sequence length increases, accessing the KV cache incurs significant overhead, thereby impacting the end-to-end latency of inference. Besides, as the sequence length increases, the KV cache itself consumes more memory space; for instance, with just a context length of 16k, a batch size of 8 nearly saturates the

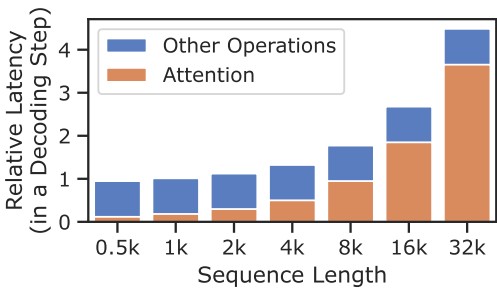 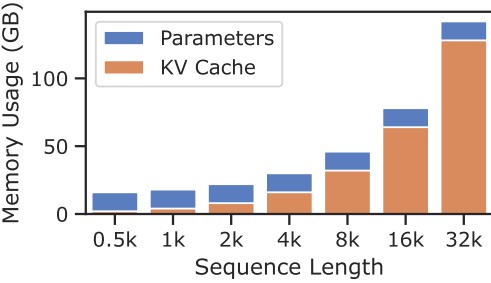

(a) Inference latency of attention and other operations.    (b) Memory usage of KV cache and parameters.

Figure 2: Latency & memory breakdown of a decoding step of LongChat-7b-v1.5-32k model with batch-size $b = 8$ and different history sequence-lengths $s = 2^9, 2^{10}, 2^{11}, 2^{12}, 2^{13}, 2^{14}, 2^{15}$.

80GB memory of an A100 GPU. This not only hampers the use of larger batch sizes but also poses challenges for deploying models with more parameters.

## 4 Observation

### 4.1 Token-level Sparsity of KV Cache

To tackle the aforementioned issues, we aim to exploit the sparsity of the KV cache. We collect data by running LongChat-7b-v1.5-32k [1] on LongBench [9]. Figure 3 illustrates the sparsity of the KV caches of different layers. We organize the tokens in KV cache into pages (with page-size=32), and rank the pages of each layer based on the highest attention scores of the tokens within each page for each decoding step. We can observe that, in each layer except the initial one, less than 10 pages (320 tokens) contribute more than 99% scores in sum, achieving over 95% sparsity. Thus we can retain only a subset of KV tokens for attention.

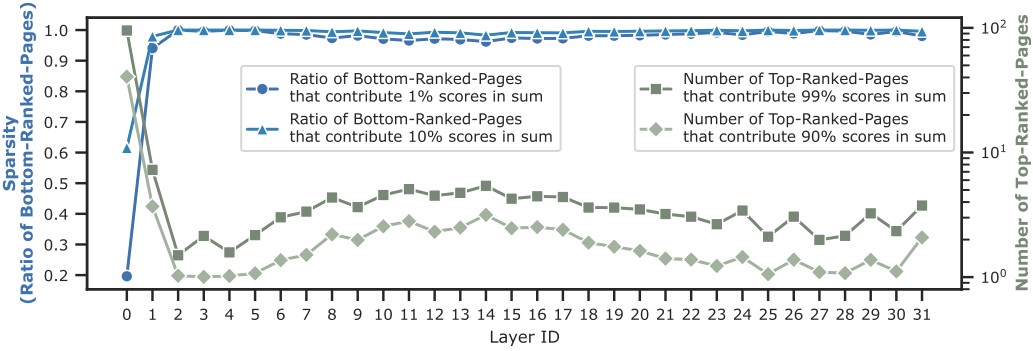

Figure 3: Token-level sparsity of KV cache. We organize the tokens in KV cache into pages (with page-size=32) and rank the pages of each layer based on the highest attention (softmax) scores of the tokens within each page.

### 4.2 Dynamism of Token Importance

Previous efforts leverage the sparsity of KV cache to control its size by retaining recent tokens while discarding older ones or selectively evicting less important tokens based on their historical attention scores, shown in Figure 1 (b) (c). However, we find that the importance of tokens in the KV cache can dynamically change over time, as shown in Figure 4a. For example, page 256 initially holds little importance but later becomes crucial, with its significant role (at position 12620) occurring nearly 4500 tokens after its sequence position (at position 8192). Previous methods risk prematurely discarding such pages to save KV cache space. To address this, we propose ARKVALE, a shown

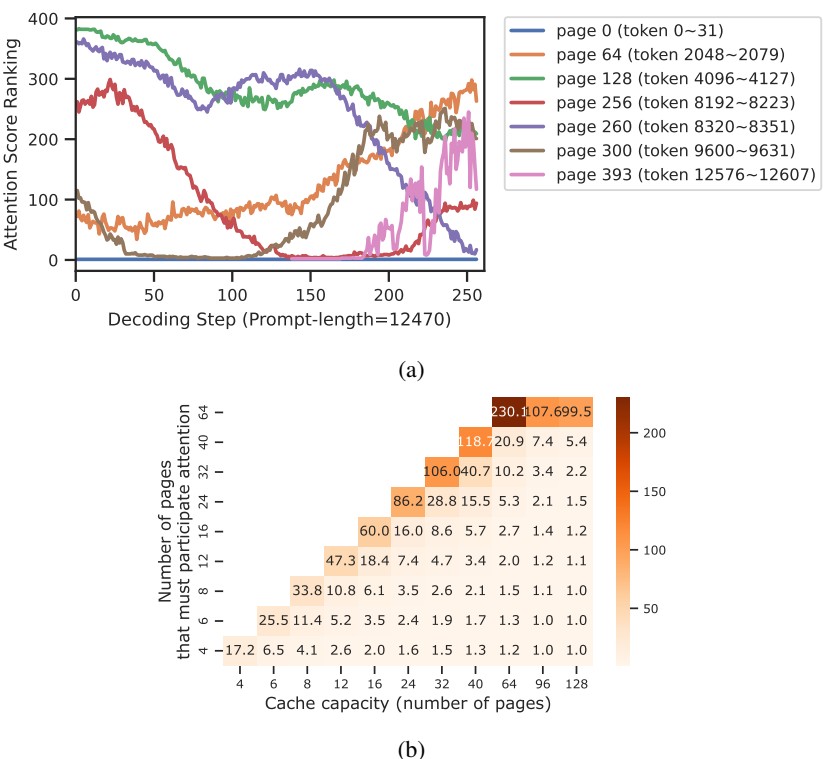

(a)

(b)

Figure 4: (a) Dynamsim of token (page) importance (page-size=32). The sample is from GovReport [28]. (b) Number of page-recalls (in average) needed during a decoding step with page-size=32.

in Figure 1 (d), a method designed to recognize shifts in token importance and properly recall vital tokens as well as evict unimportant ones, thereby preventing substantial drops in accuracy.

## 5  Techniques of ARKVALE

### 5.1  System Design

Figure 5 shows the design of ARKVALE. Inspired by vLLM [36], ARKVALE arranges KV cache tokens into pages, enabling coarse-grained management of tokens. As shown in Figure 5 (a), once a page is filled, its keys & values are asynchronously moved from GPU to external memory (CPU memory typically) for backup, with keys summarized into a digest kept on the GPU (explained in § 5.2). Prior to attention computation, ARKVALE gauges the importance of each page (whether cached or evicted), using the query and page digests (Figure 5 (b), detailed in § 5.2), and subsequently ranks these pages (Figure 5 (c)) based on their importance scores. The top-$k$ pages (governed by a hyper-parameter $k$) must engage in attention computation (Figure 5 (e)). If any top-$k$ pages were evicted before, they will be recalled from backup, and bottom-ranked pages in GPU will be evicted for exchange (Figure 5 (d)).

Depending on the cache-size and the number of pages selected for attention, the overhead of page recall may be different. To validate the feasibility of our approach and ensure that recalls do not incur excessive overhead, we uniformly sample four instances from each dataset included in LongBench [9] and gather data from the inference process of LongChat-7b-v1.5-32k [1] to simulate eviction and recall scenarios. We configure the page-size $p = 32$ and vary both the cache capacity $c$ (pages) and the attention size $k$ (the number of top-ranked pages selected for attention). The results, depicted in Figure 4b, reveal that as long as $k < \min(40, c/2)$, we can limit the number of recalls per decoding step to under 10, amounting to a total of 5 MB of data transfer. This transfer takes merely hundreds of microseconds over contemporary CPU-GPU communication interfaces, which is one or two orders of magnitude faster than the end-to-end latency of each decoding step (usually a few

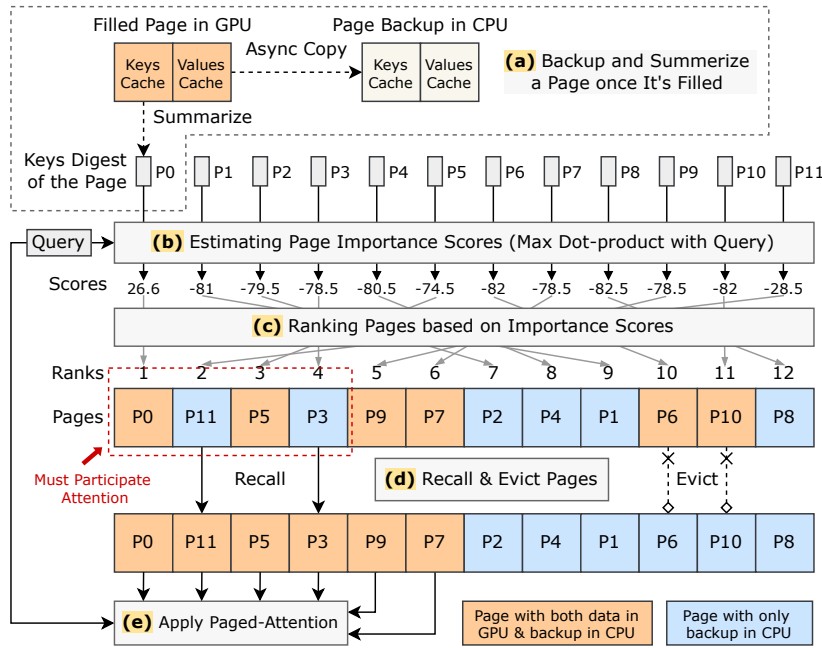

Figure 5: Design overview of ARKVALE

to tens of milliseconds in long context scenarios on NVIDIA A100 GPU). Thus, for page-size $p$ and cache-capacity $c$ (tokens), we set $k = \min(C, c/2)/p$, where $C$ is a hyper-parameters (default $C = 40 * 32 = 1280$).

## 5.2 Page Summarization & Importance Estimation

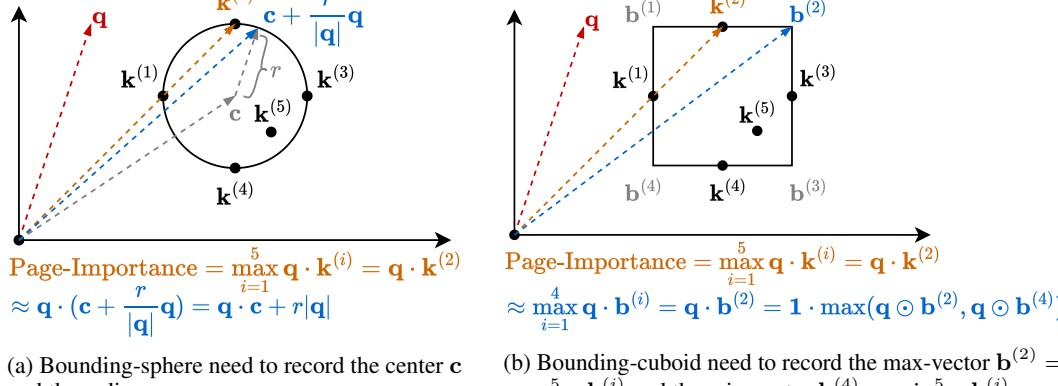

Page-Importance $= \max_{i=1}^{5} \mathbf{q} \cdot \mathbf{k}^{(i)} = \mathbf{q} \cdot \mathbf{k}^{(2)}$
$\approx \mathbf{q} \cdot (\mathbf{c} + \frac{r}{|\mathbf{q}|}\mathbf{q}) = \mathbf{q} \cdot \mathbf{c} + r|\mathbf{q}|$

(a) Bounding-sphere need to record the center $\mathbf{c}$ and the radius $r$.

Page-Importance $= \max_{i=1}^{5} \mathbf{q} \cdot \mathbf{k}^{(i)} = \mathbf{q} \cdot \mathbf{k}^{(2)}$
$\approx \max_{i=1}^{4} \mathbf{q} \cdot \mathbf{b}^{(i)} = \mathbf{q} \cdot \mathbf{b}^{(2)} = \mathbf{1} \cdot \max(\mathbf{q} \odot \mathbf{b}^{(2)}, \mathbf{q} \odot \mathbf{b}^{(4)})$

(b) Bounding-cuboid need to record the max-vector $\mathbf{b}^{(2)} = \max_{i=1}^{5} \mathbf{k}^{(i)}$ and the min-vector $\mathbf{b}^{(4)} = \min_{i=1}^{5} \mathbf{k}^{(i)}$.

Figure 6: Summarize page keys $\{\mathbf{k}^{(i)}\}_{i=1}^{5}$ into their bounding-volume (sphere/cuboid). We can estimate the max-dot-product between query $\mathbf{q}$ and keys $\{\mathbf{k}^{(i)}\}_{i=1}^{5}$ using the bounding-volume.

For every filled page, we maintain a digest as a compression of it. The digest is much smaller than the page itself, and can be used to estimate the importance of each page regardless of whether the page has been evicted before attention computation. This estimation enables us to rank pages to selectively recall or evict pages based on their importance.

Given a page with $n$ keys $K = \{\mathbf{k}^{(i)}\}_{i=1}^{n}$ ($\mathbf{k}^{(i)} \in \mathbb{R}^d$), and a query $\mathbf{q} \in \mathbb{R}^d$, we measure its importance using the maximum dot product between the query and the keys: $I(\mathbf{q}, K) = \max_{\mathbf{k} \in K} \mathbf{q} \cdot \mathbf{k}$. For two sets of keys, $K_1$ and $K_2$, $I(\mathbf{q}, K_1) > I(\mathbf{q}, K_2)$ implies that there's at least one token in $K_1$ with higher attention (softmax) scores compared to every token in $K_2$.

It's evident that $\text{argmax}_{\mathbf{k} \in K} \mathbf{q} \cdot \mathbf{k}$ must lie on a vertex of the convex hull of the point set $K$. Hence, we can approximate $I(\mathbf{q}, K)$ by constructing a convex covering set of $K$. We leverage the concept of **bounding-volume** [3, 34] to summarize the keys $K$ of a page. A bounding-volume for a set of points is a closed region that completely contains all of them, which is widely used in computer graphics [34, 29, 11, 12].

**Bounding-sphere.** We can cover the keys $K$ with a sphere, needing to store only a center $\mathbf{c} \in \mathbb{R}^d$ and a radius $r \in \mathbb{R}$ as the digest. Given a query $\mathbf{q} \in \mathbb{R}^d$, we can estimate the importance $I(\mathbf{q}, K)$ leveraging $\mathbf{c}$ and $r$. Among the points $\mathbf{k}'$ on the surface of the sphere, those for which $\mathbf{k}' - \mathbf{c}$ is parallel to $\mathbf{q}$ yield the maximum dot product with $\mathbf{q}$. Thus we have $\mathbf{k}' = \frac{r}{|\mathbf{q}|}\mathbf{q}$, leading to:

$$I(\mathbf{q}, K) = \max_{\mathbf{k} \in K} \mathbf{q} \cdot \mathbf{k} \approx \mathbf{q} \cdot (\mathbf{c} + \frac{r}{|\mathbf{q}|}\mathbf{q}) = \mathbf{q} \cdot \mathbf{c} + r|\mathbf{q}| \tag{1}$$

An example is shown in Figure 6a. Finding the minimal bounding sphere is complicated; hence, we employ an efficient approximation method: we take the center of the Axis-Aligned Bounding Box (AABB) [12, 11] of $K$ as our sphere center: $\mathbf{c} = \frac{1}{2}(\min_{\mathbf{k} \in K} \mathbf{k} + \max_{\mathbf{k} \in K} \mathbf{k})$ (min and max are element-wise operations in this paper). For the radius $r$, three alternatives are considered:

$$r_{\max} = \max_{\mathbf{k} \in K} |\mathbf{c} - \mathbf{k}| \quad r_{\text{center}} = \frac{1}{2}(\min_{\mathbf{k} \in K} |\mathbf{c} - \mathbf{k}| + r_{\max}) \quad r_{\text{mean}} = \frac{1}{|K|}\sum_{\mathbf{k} \in K} |\mathbf{c} - \mathbf{k}| \tag{2}$$

where $|...|$ means L2-norm. Strictly speaking, only $r_{\max}$ guarantees to enclose all points in $K$, but the latter two can avoid overestimating $I(\mathbf{q}, K)$ in practice, as shown in Figure 7 and discussed in §6.2.

**Bounding-cuboid.** We can also cover the keys $K$ directly using an Axis-Aligned Bounding Box (AABB) [12, 11]. It needs to store the max-vector $\mathbf{b}^{\max} = \max_{\mathbf{k} \in K} \mathbf{k}$ and min-vector $\mathbf{b}^{\min} = \min_{\mathbf{k} \in K} \mathbf{k}$ as the digest. Given a query vector $\mathbf{q} \in \mathbb{R}^d$, we can estimate the importance $I(\mathbf{q}, K)$ of $K$ with the boundaries of the box. This is achieved by summing the maximum product by the boundary values and $\mathbf{q}$ across all dimensions:

$$I(\mathbf{q}, K) = \max_{\mathbf{k} \in K} \mathbf{q} \cdot \mathbf{k} \approx \sum_{i=1}^{d} \max_{\mathbf{k} \in K} \mathbf{q}_i \mathbf{k}_i = \sum_{i=1}^{d} \max(\mathbf{q}_i \mathbf{b}_i^{\max}, \mathbf{q}_i \mathbf{b}_i^{\min}) = \mathbf{1} \cdot \max(\mathbf{q} \odot \mathbf{b}^{\max}, \mathbf{q} \odot \mathbf{b}^{\min})$$
$$\tag{3}$$

Here, $\mathbf{1} \in \mathbb{R}^d$ denotes a vector of ones, and $\odot$ means element-wise multiplication. An example is shown in Figure 6b. Similar to the bounding-sphere approach, we can employ different "radius vectors" to define cuboids with varying sizes:

$$\mathbf{r}^{\max} = \max_{\mathbf{k} \in K} \text{abs}(\mathbf{c} - \mathbf{k}) \quad \mathbf{r}^{\text{center}} = \frac{1}{2}(\min_{\mathbf{k} \in K} \text{abs}(\mathbf{c} - \mathbf{k}) + \mathbf{r}^{\max}) \quad \mathbf{r}^{\text{mean}} = \frac{1}{|K|}\sum_{\mathbf{k} \in K} \text{abs}(\mathbf{c} - \mathbf{k}) \tag{4}$$

where $\text{abs}(...)$ computes element-wise absolute values. For each $\mathbf{r}$, we can derive $\mathbf{b}^{\max} = \mathbf{c} + \mathbf{r}$ and $\mathbf{b}^{\min} = \mathbf{c} - \mathbf{r}$, where $\mathbf{c} = \frac{1}{2}(\min_{\mathbf{k} \in K} \mathbf{k} + \max_{\mathbf{k} \in K} \mathbf{k})$ is the center of the AABB of $K$.

## 6 Evaluation

### 6.1 Experimental Setup

We apply our method to LongChat-7b-v1.5-32k [1] and use 6 datasets from LongBench [9] for benchmarking: HotpotQA [59], NarrativeQA [35], Qasper [20], GovReport [28], TriviaQA [30], and PassageRetrieval [9], along with the passkey-retrieval tasks. For comparison, we choose the state-of-the-art KV cache eviction methods including StreamingLLM [58], H2O [64], and TOVA [40] as baselines. As Figure 3 illustrates, the initial layers exhibit relatively low sparsity; hence, we do not apply ARKVALE and baselines to the first two layers of model.

Our experiment platform comprises an Intel(R) Xeon(R) Gold 6348 CPU @ 2.60GHz and an NVIDIA A100 80GB PCIe GPU. The software stack includes CUDA version 12.3, PyTorch [41, 8] version 2.3.0, and HuggingFace Transformers [57] version 4.40.0. We implement ARKVALE on top of Huggingface Transformers, with CUTLASS [54], FlashInfer [60], and RAFT [44] for certain kernels.

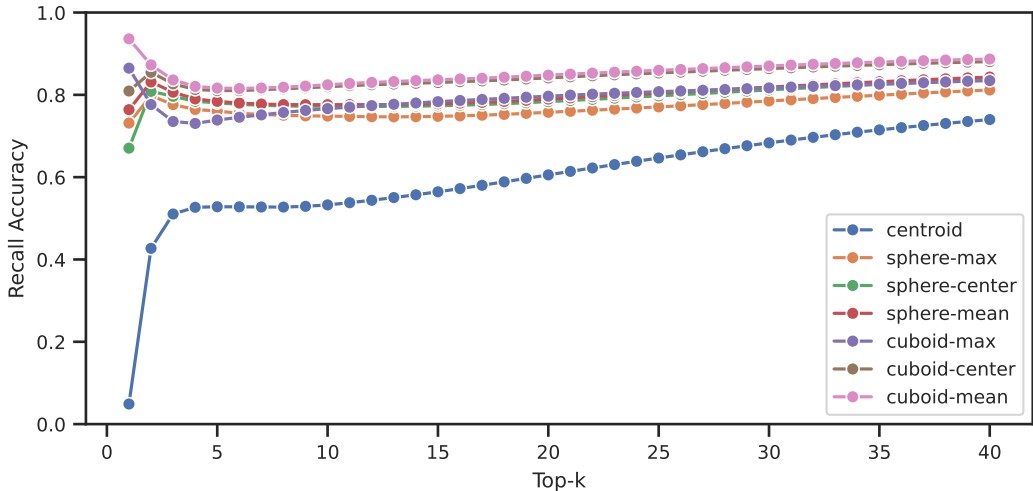

Figure 7: Recall accuracy (the proportion of pages predicted to be among the top-k that indeed belong to the top-k) of different estimation methods. "Centroid" is the baseline which just uses the centroid of keys to estimate the max-dot-product with given query.

## 6.2 Estimation Accuracy

We begin by evaluating the accuracy of the page summarization & importance estimation methods discussed in §5.2. Using data from the LongBench [9] datasets employed in our experiments, we simulate each decoding step to collect both the actual page rankings and the estimated rankings derived using the digests. For varying values of $k$, we define recall accuracy as the proportion of overlap between the estimated top-$k$ page set $E_k$ and the true top-$k$ page set $R_k$, that is: $|E_k \cap R_k|/|R_k|$.

Figure 7 illustrates the recall accuracy of different estimation techniques across various $k$ values. The "centroid" method, which uses the centroid [4] (element-wise average of keys) of page keys as the digest and estimates page importance by directly taking the dot product of the query and the centroid, serves as a straightforward baseline. Other methods are the six introduced in §5.2: spheres with $r_{\max}$, $r_{\text{center}}$, and $r_{\text{mean}}$ as well as cuboids with $\mathbf{r}^{\max}$, $\mathbf{r}^{\text{center}}$, and $\mathbf{r}^{\text{mean}}$. As shown in Figure 7, the centroid method cannot recall the top-4 pages with even 50% accuracy and achieves less than 5% accuracy in recalling the top-1 page, substantially undermining inference accuracy. Conversely, our six proposed methods guarantee at least a 60% recall accuracy, with the standout cuboid-mean method ensuring a 95% accuracy for top-1 recall and consistently over 80% for other $k$ values. The cuboid-based methods generally outperform their sphere-based counterparts due to more retained information (two vectors for cuboids versus one vector and a scalar for spheres). Furthermore, the "mean" variants tend to perform better than others, potentially because they provide a more balanced boundary estimation, avoiding overestimation of page importance.

## 6.3 LongBench Results

Next, we evaluate the accuracy of ARKVALE in general long-context tasks. We employ 6 datasets from LongBench [9] as benchmarks, covering tasks such as multi-document QA with HotpotQA [59], single-document QA with NarrativeQA [35], and Qasper [20], text summarization with GovReport [28], few-shot learning with TriviaQA [30], and synthetic tasks with PassageRetrieval [9]. The target model is LongChat-7b-v1.5-32k [1]. Our comparison baselines include the original model and the state-of-the-art KV cache eviction methods: StreamingLLM [58], H2O [64], and TOVA [40]. We configure four cache budget settings: 4096, 2048, 1024, and 512. Notably, H2O and TOVA's original implementations require the full attention scores matrix to maintain states, preventing the use of optimizations like Flash-Attention [19] in prefill stage and causing Out-of-Memory (OOM) for many benchmark samples. To address this, we split each input into context and question/instruction sections and adopt a two-phase prefill: the first phase uses Flash-Attention [19] to handle the lengthy context without OOM (while we also separately compute the last row of attention scores for H2O

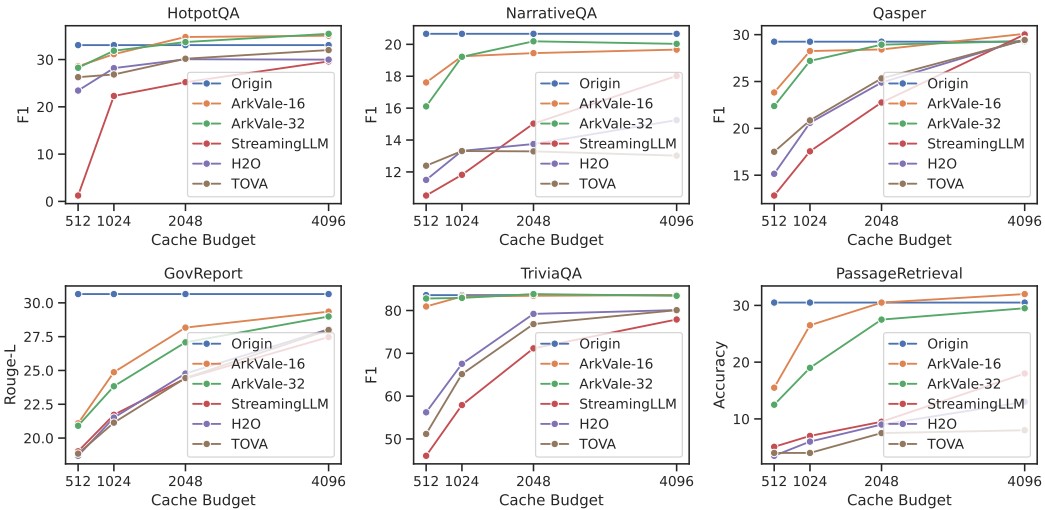

Figure 8: Evaluation on 6 long-context datasets in LongBench [9] with different cache budgets.

and TOVA's initial states update with little memory overhead), and the second phase uses brutal-force attention to process the shorter instruction, ensuring a complete score matrix for H2O and TOVA's states updates. This approach can also better demonstrate the long-range dependencies handling abilities of each method.

Figure 8 shows the results, where ARKVALE-16 and ARKVALE-32 represent ARKVALE with page-sizes of 16 and 32, respectively. ARKVALE persistently surpasses baselines across datasets and cache budgets. It nearly equals Origin's performance with budgets over 2048, while other baselines show noticeable disparities when the budget dips below 2048 and even 4096. Specifically, ARKVALE achieves comparable results to Origin at budgets of 1024 for HotpotQA, 2048 for NarrativeQA and PassageRetrieval, 1024 for Qasper, and 512 for TriviaQA. Page-size differences have little impact in most test-cases; however, ARKVALE-16 often outperforms ARKVALE-32 when budgets are tight, as it selects more pages under the same limited budget, enhancing the chances of hitting vital tokens.

## 6.4 Passkey Retrieval Results

Table 1: Accuracy of passkey retrieval tasks

| Context Length | 10k | | | | 20k | | | | 30k | | | |
|---|---|---|---|---|---|---|---|---|---|---|---|---|
| Cache Budget | 512 | 1024 | 2048 | 4096 | 512 | 1024 | 2048 | 4096 | 512 | 1024 | 2048 | 4096 |
| StreamingLLM [58] | 0% | 5% | 15% | 40% | 0% | 0% | 5% | 20% | 0% | 0% | 5% | 10% |
| H2O [64] | 5% | 5% | 15% | 40% | 0% | 5% | 5% | 20% | 5% | 5% | 5% | 15% |
| TOVA [40] | 5% | 10% | 20% | 40% | 5% | 5% | 10% | 20% | 5% | 5% | 5% | 15% |
| **ARKVALE** | **100%** | **95%** | **100%** | **95%** | **95%** | **100%** | **100%** | **100%** | **100%** | **95%** | **95%** | **100%** |

We further examine the accuracy of ARKVALE under long-range dependency scenarios in detail. We employ the passkey retrieval task [39] as our benchmark. We establish three context lengths for our tests: 10k, 20k, and 30k. For each context length, we generate 20 unique test cases, each with the passkey inserted at depths corresponding to 0%, 5%, ..., up to 95% of the text's length. Following the setup in Section §6.3, we compare ARKVALE against StreamingLLM [58], H2O [64], and TOVA [40]. Similarly, we adopt two-phase prefill for input texts to prevent OOM incidents of H2O and TOVA and more effectively evaluate each method's long-range dependencies handling.

The results are shown in Table 1. Baselines, which permanently evict some tokens, risk discarding passkey-related information during inference, leading to accuracy falling short of 50% and declining further as the context length increases or the cache budget decreases. In contrast, ARKVALE examines the importance of evicted tokens (pages) and promptly recalls vital ones, thereby maintaining a stable passkey retrieval accuracy above 95% across all tested context lengths and cache budgets.

## 6.5 Performance Results

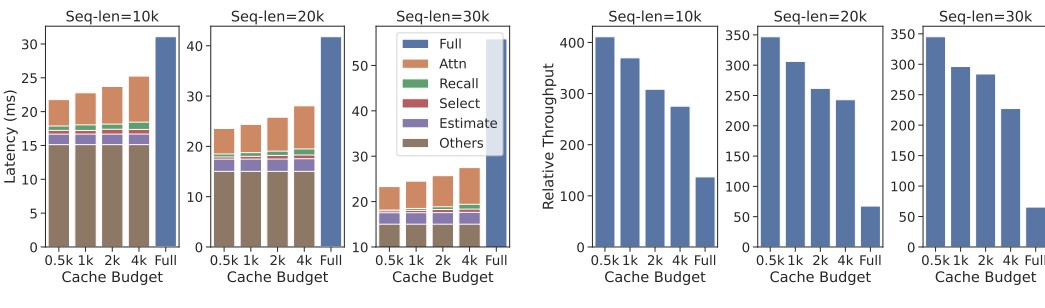

(a) Latency Breakdown (batch-size=4)  (b) Throughput Comparison

Figure 9: Decoding latency breakdown and achievable maximum throughput with page-size=32 and different seq-lens & cache budgets compared to model with full KV cache.

We set up test-cases with lengths close to 10k, 20k, and 30k from GovReport [28] in LongBench [9] and examined ARKVALE's inference latency with settings of a batch-size=4, page-size=32, and KV cache budgets of 512, 1024, 2048, and 4096. Figure 9a shows the results. Here, "Full" refers to the baseline without KV cache eviction, whereas "Estimate," "Select," and "Recall" denote the overhead for estimating page importance, choosing pages for computations and evictions, and recalling pages from CPU memory, respectively. ARKVALE outperforms the baseline across all the tested text lengths, reaching up to $2.2\times$ speedup ($1.7\times$ in average). The main bottleneck of the baseline is the full-attention, whose latency grows linearly with seq-len, whereas our attention's latency is mainly governed by the fixed cache budget. Our extra latency overhead mostly arises from estimating importance, which grows linearly with the $\#pages = \frac{\text{seq-len}}{\text{page-size}}$. Recalling, conversely, adds small overhead (especially when seq-len is large and cache-budget is small), consistent with the discussion in §5.1.

We further compare the achievable maximum throughput in serving scenarios of ARKVALE versus the baseline under a 40 GB memory limit for KV cache. Figure 9b shows the results. Our throughput reaches up to $4.6\times$ ($3.5\times$ in average) to that of baseline. This achievement is not only due to the reduced inference latency but also to a decreased per sample memory usage of the KV cache, enabling larger batch-size to enhance weight sharing to lessen the average latency of Linear computations per sample.

## 7 Limitation

The limitation of ARKVALE mainly lies on storing a backup for each KV cache page in external memory (typically the CPU memory). Although the latency of data transfer for backup during decoding phase can be hided by asynchronous copy, the backup latency during prefilling phase is hard to completely overlapped. Also, the backups may occupy much CPU memory. When the CPU's memory capacity is insufficient, these backups may need to be offloaded to disk storage.

## 8 Conclusion

In this paper, we propose ARKVALE, a method designed for managing KV cache eviction and recall. It organizes KV cache tokens into pages, backs them up, and creates summaries, enabling the recognition of page importance to recall vital pages evicted before. Our method performs well on various long context tasks with few accuracy loss under a cache budget of 2k∼4k and speeds up decoding latency up to $2.2\times$ and boosts decoding throughput up to $4.6\times$ in long-context scenarios.

## 9 Acknowledgments

This work is supported in part by the National Natural Science Foundation of China (No. T2325001) and Beijing Municipal Science and Technology Program (No. Z241100004224015).

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

# A Appendix / supplemental material

Optionally include supplemental material (complete proofs, additional experiments and plots) in appendix. All such materials **SHOULD be included in the main submission.**

