# OpenReview forum: "ArkVale: Efficient Generative LLM Inference with Recallable Key-Value Eviction"
_NeurIPS.cc/2024/Conference — NeurIPS 2024 poster_

### Official Review · Reviewer_YdSG · 2024-06-27

**Soundness:** 3
**Presentation:** 4
**Contribution:** 3
**Rating:** 7
**Confidence:** 4

**Summary:**

This paper proposes a paged-based KV cache manager that identifies and recalls important tokens for LLM inference, termed ArkVale. Results show that ArkVale achieves 2.2x latency and 4.6x throughput improvement on various long context tasks.

**Strengths:**

1. The paper is easy to follow, with clear writing and presentation.
2. Evaluation results are comprehensive.

**Weaknesses:**

1. How does the page size affect the memory consumption for the KV cache? Would smaller page size lead to potential fragmentation issues?
2. In the related work section, it would be nice if the authors could the relationship between ArkVale and some concurrent works [1,2,3].

[1] Keyformer: KV Cache Reduction through Key Tokens Selection for Efficient Generative Inference, MLSys 2024.

[2] Q-Hitter: A Better Token Oracle for Efficient LLM Inference via Sparse-Quantized KV Cache, MLSys 2024.

[3] ALISA: Accelerating Large Language Model Inference via Sparsity-Aware KV Caching, ISCA 2024.

**Questions:**

Please see the weaknesses above.

**Limitations:**

Please see the weaknesses above.

---

> ### Author Rebuttal · Authors · 2024-08-06
>
> >How does the page size affect the memory consumption for the KV cache? Would smaller page size lead to potential fragmentation issues?
>
> We allocate pages from a pre-allocated memory pool, and all pages have the same page-size, thus avoiding the issue of memory fragmentation. However, a smaller page-size results in an increased number of pages, leading to a greater number of page digests and consequently increased memory usage.
>
> ---
>
> >In the related work section, it would be nice if the authors could the relationship between ArkVale and some concurrent works [1,2,3].
>
> Keyformer [1] are similar to works like H2O in that they use historical information to perform eviction on the KV-Cache, and they have improved the eviction score function by leveraging the Gumbel distribution. Q-Hitter [2] combines KV-cache quantization with KV-cache eviction. But neither of these works can dynamically assesses the importance of tokens.
>
> ALISA [3] uses a post-training dynamic sparse attention approach, similar to SparQ, which dynamically evaluates the importance of each token in the KV-cache and selects a subset of tokens for attention calculation. However, this token-level approach is too granular and incurs significant additional overhead (requiring all KV-cache tokens to be involved in assessing importance), and it also poses challenges for memory management if it requires token-level eviction & recall. In contrast, our approach operates at the page-level for importance estimation, eviction, and recall, achieving a good balance between accuracy and performance overhead.
>
> [1] Keyformer: KV Cache Reduction through Key Tokens Selection for Efficient Generative Inference, MLSys 2024.
>
> [2] Q-Hitter: A Better Token Oracle for Efficient LLM Inference via Sparse-Quantized KV Cache, MLSys 2024.
>
> [3] ALISA: Accelerating Large Language Model Inference via Sparsity-Aware KV Caching, ISCA 2024.

---

> ### Comment · Reviewer_YdSG · 2024-08-07
>
> Thank you for your detailed response. I don't have any further questions.
>
> My only suggestion would be to incorporate the discussion of concurrent works into the final version if the paper is accepted.

---

### Official Review · Reviewer_8mmm · 2024-07-12

**Soundness:** 3
**Presentation:** 3
**Contribution:** 3
**Rating:** 6
**Confidence:** 4

**Summary:**

ARKVALE presents a page-based key-value (KV) cache manager designed to address the challenges associated with long-context processing in large language models. The main contribution is its ability to recognize and recall important tokens that were previously evicted, thereby optimizing memory usage and improving throughput and latency without significant accuracy loss. The method involves asynchronously backing up filled pages in external memory and using bounding-volume techniques to summarize and estimate the importance of these pages for efficient recall and eviction. Experiments show strong performance in terms of both accuracy and efficiency.

**Strengths:**

- ARKVALE empirically identifies the dynamism of token importance and proposes efficient token eviction and recall methods. This approach ensures good memory efficiency while maintaining accuracy.
- The efficient system design and implementation improve decoding latency by up to 2.2x and batching throughput by up to 4.6x.
- ARKVALE performs well across different tasks by comprehensive benchmarking, shows its ability in various long-context scenarios.

**Weaknesses:**

- The benchmark is only conducted on LongChat, lacking evaluations on different models such as Mistral and LLaMA-3.
- There is a lack of comparison with other methods focusing on efficiency, and some other baselines like [1,2].
- The paper lacks discussion and ablation studies on some hyper-parameters, such as top-k, page-size and the relationship between top-k, page-size and the cache budget.

[1]  Infllm: Unveiling the intrinsic capacity of llms for understanding extremely long sequences with training-free memory

[2] Snapkv: Llm knows what you are looking for before generation

**Questions:**

- What is the performance of ARKVALE on Mistral-7B-Instruct-v0.2 and LLaMA-3-8B-Instruct?
- How does the performance (accuracy, latency/throughput) of ARKVALE compare to other methods [1, 2]?
- What is the process for selecting the hyperparameters, and what their influence on performance?
- Does the prefill phase use all tokens? For example, for the first token generated, is full attention used? Additionally, is this method compatible with FlashAttention/FlashDecoding?

[1]  Infllm: Unveiling the intrinsic capacity of llms for understanding extremely long sequences with training-free memory

[2] Snapkv: Llm knows what you are looking for before generation

**Limitations:**

See weaknesses and questions.

---

> ### Author Rebuttal · Authors · 2024-08-06
>
> >What is the performance of ARKVALE on Mistral-7B-Instruct-v0.2 and LLaMA-3-8B-Instruct?
>
> We conduct experiments on our method adapted to Mistral-7B and Llama-3-8B, as detailed in Global Response.
>
> ---
>
> >How does the performance (accuracy, latency/throughput) of ARKVALE compare to other methods
>
> In the Global Response, we compare the accuracy and latency of ArkVale with other baselines.
>
> ---
>
> >The paper lacks discussion and ablation studies on some hyper-parameters, such as top-k, page-size and the relationship between top-k, page-size and the cache budget.
> >What is the process for selecting the hyperparameters, and what their influence on performance?
>
> For the cache-budget $c$ and page-size $p$, we empirically set the top-k to $\min(40*32, c/2) / p$. In the Global Response, we discuss the impact of different cache-budgets and page-sizes on accuracy and latency.
>
> ---
>
> >Does the prefill phase use all tokens? For example, for the first token generated, is full attention used?
>
> Yes, currently we use the full kv-cache during the prefill phase. However, it is also feasible to perform eviction early in the prefill phase, although the impact on accuracy needs to be determined through experiments. We plan to explore this in future work.
>
> ---
>
> >Additionally, is this method compatible with FlashAttention/FlashDecoding?
>
> Yes, but it requires using the Paged version of FlashAttention/FlashDecoding, such as the kernels implemented in FlashInfer.

---

> ### Comment · Reviewer_8mmm · 2024-08-12
>
> Thanks for the reply, this answers my questions.

---

### Official Review · Reviewer_ntoN · 2024-07-15

**Soundness:** 3
**Presentation:** 3
**Contribution:** 3
**Rating:** 7
**Confidence:** 5

**Summary:**

The paper proposed a method to minimize the risk of KV cache eviction by efficiently and soundly offloading some of them into external memory, which is realized by page organization, page digest, and digest ranking/scoring. The method gets much better performance in context retrieval tasks compared to other KV eviction methods.

**Strengths:**

The paper proposes a reliable way to do KV cache eviction with minimal risk.
The system is sound and intuitive. It seems compatible with real world frameworks like vLLM.
Experiments show good results.

**Weaknesses:**

It's not very clear if the dynamics of importance will change along the decoding in other tasks, so the observations may be limited.
It would be interesting to see how the page size affect the methods, as vLLM's default page size is 16. Is there any reason to scale to 32?
Would the method still work when TP>1?

**Questions:**

I'm wondering if there will be an open-sourced implementation/PR to vLLM?

**Limitations:**

It's desirable to see normal length tasks, e.g., GSM8K.

---

> ### Author Rebuttal · Authors · 2024-08-06
>
> > It would be interesting to see how the page size affect the methods, as vLLM's default page size is 16. Is there any reason to scale to 32?
>
> We briefly discuss the impact of different page-sizes on accuracy and latency in Global Response. A page-size of 32 is a compromise between these two aspects (though in fact, a page-size of 16 would also be perfectly acceptable).
>
> ---
>
> >Would the method still work when TP>1?
>
> Yes, ArkVale can be applied to distributed scenarios. But in the context of distributed inference for LLMs, it is common for different heads of a KV-cache to be distributed across various GPUs. Thus, head-wise page eviction and recall may be necessary.
>
> ---
>
> >I'm wondering if there will be an open-sourced implementation/PR to vLLM?
>
> Yes, we plan to introduce serving technologies such as continuous batching into ArkVale and adapt it to serving frameworks like vLLM and SGLang.
>
> ---
>
> >It's desirable to see normal length tasks, e.g., GSM8K.
>
>
> We evaluate our method on six normal length benchmarks using `lm-eval-harness`. The results are presented in the table below (with page-size=32):
>
> | Cache Budget        | \|  | Full  | 4096  | 2048  | 1024  | 512   |
> | ------------------- | --- | ----- | ----- | ----- | ----- | ----- |
> | GSM8K (5-shot)      | \|  | 0.092 | 0.092 | 0.092 | 0.086 | 0.078 |
> | HellaSwag (0-shot)  | \|  | 0.544 | 0.544 | 0.544 | 0.544 | 0.544 |
> | WinoGrande (0-shot) | \|  | 0.684 | 0.683 | 0.683 | 0.683 | 0.683 |
> | PIQA (0-shot)       | \|  | 0.760 | 0.761 | 0.761 | 0.761 | 0.761 |
> | OpenBookQA (0-shot) | \|  | 0.306 | 0.306 | 0.306 | 0.306 | 0.306 |
> | MathQA (0-shot)     | \|  | 0.253 | 0.253 | 0.253 | 0.253 | 0.253 |
>
> The data from the table indicates that, in most cases, ArkVale maintains comparable accuracy to the original model when the cache budget does not exceed 4096. This is likely because the context lengths for these tasks are not long enough, and they primarily test the model's fundamental capabilities rather than its contextual memory. However, in tasks like GSM8K, a notably small cache budget can lead to an obvious drop in accuracy.

---

### Official Review · Reviewer_PLAC · 2024-07-18

**Soundness:** 3
**Presentation:** 3
**Contribution:** 3
**Rating:** 6
**Confidence:** 4

**Summary:**

The paper introduces ARKVALE, a novel page-based key-value (KV) cache management approach designed to optimize the performance of Large Language Models (LLMs) when dealing with long context lengths. As the demand for higher context lengths in tasks such as multi-turn chats and content generation increases, the management of extended key-value caches becomes crucial due to memory constraints and the impact on computation latency and throughput. ARKVALE addresses these challenges by dynamically managing the cache, selectively evicting less important tokens while recalling those that become relevant again at different decoding steps. This strategy leverages a mechanism that organizes tokens into pages and uses digests to estimate the importance of these pages. By summarizing pages into smaller, more manageable units, ARKVALE efficiently decides which pages to recall from external memory and which to evict, enabling focused attention computations on only the most relevant subsets of data.

The paper's experiments demonstrate that ARKVALE effectively handles various long-context tasks with minimal loss in accuracy, even under constrained cache sizes between 2K and 4K tokens. The results indicate significant improvements in model decoding latency and batching throughput, specifically up to 2.2x faster latency and 4.6x higher throughput. These enhancements are achieved by applying attention to a reduced subset of pages, thus decreasing the per-sample memory usage of the KV cache. This system not only improves the operational efficiency of LLMs but also maintains high accuracy levels, suggesting a scalable solution for managing extensive data in real-time language processing applications.

**Strengths:**

**Originality**: ARKVALE introduces a novel page-based system for KV cache management that dynamically adjusts which data is retained or discarded based on the evolving importance of tokens throughout the decoding process. This approach creatively combines ideas from memory management and attention mechanisms in LLMs, setting it apart from previous methods that often permanently discard tokens without the ability to recall them.

**Quality**: The methods proposed are rigorously tested across various benchmarks and scenarios, demonstrating minimal loss in accuracy while substantially improving efficiency in terms of decoding latency and throughput. The experimental setup is thorough, using several datasets to ensure robustness and reproducibility of results.

**Clarity**: The paper is well-organized, presenting complex ideas in a structured and understandable manner. The use of diagrams to illustrate the KV cache management process helps in demystifying the approach and makes the operational details of ARKVALE accessible to readers.

**Significance**: ARKVALE's impact is multifaceted. For practical applications, it allows for the deployment of LLMs in environments where memory and latency are constraints, thus broadening their usability in real-world applications. Theoretically, it advances our understanding of efficient memory management in models requiring extensive context, potentially influencing future developments in LLM architectures and optimization techniques.

**Weaknesses:**

- Insufficient Model Evaluation:
The paper evaluates the proposed method using only the LongChat-7b-v1.5-32k model, which is relatively outdated in the current landscape. It does not demonstrate the method's generality and robustness across different model architectures (e.g., MoE, GQA) and scales (70B models).

**Questions:**

Clarification on Model Choice: The paper primarily utilizes the LongChat-7b-v1.5-32k model for evaluating ARKVALE. Can the authors provide specific reasons for choosing this model over others? Additionally, how do the authors anticipate ARKVALE would perform with other, potentially larger or more recent LLM architectures?

**Limitations:**

The authors have not discussed the limitations of their work in the submission. It would be great to add sections discussing the limitations and societal impacts of ARKVALE.

---

> ### Author Rebuttal · Authors · 2024-08-06
>
> >Clarification on Model Choice: The paper primarily utilizes the LongChat-7b-v1.5-32k model for evaluating ARKVALE. Can the authors provide specific reasons for choosing this model over others? Additionally, how do the authors anticipate ARKVALE would perform with other, potentially larger or more recent LLM architectures?
>
> We primarily choose the LongChat-7b-v1.5-32k model because it is a relatively state-of-the-art long-text extension of Llama2 at the time.
>
> We conduct experiments on our method adapted to the relatively newer models
> Mistral-7B and Llama-3-8B that utilize GQA. The results are shown in Global Response.
>
> ---
>
> >The authors have not discussed the limitations of their work in the submission. It would be great to add sections discussing the limitations and societal impacts of ARKVALE.
>
> Chiefly, the limitation of our method lies on storing a backup for each KV cache page in external memory (typically the CPU memory). On the one hand, although the latency of data transfer for backup can be hided by asynchronous copy, the energy consumption cannot be eliminated. On the other hand, it may occupy much CPU memory, potentially impacting the performance of other applications under some extreme conditions. Furthermore, when the CPU's memory capacity is insufficient, these backups may need to be offloaded to disk storage.

---

> > ### Comment · Reviewer_PLAC · 2024-08-13
> > **Thank you for the response**
> >
> > I appreciate the authors' responses; most concerns have been addressed. I will keep my evaluation for acceptance.
> >
> > I also noticed that this paper is quite similar to a paper [1] presented at ICML 2024, published after the NeurIPS submission deadline. While not obligatory, discussing and comparing this work with that paper would be beneficial.
> >
> > [1] Quest: Query-Aware Sparsity for Efficient Long-Context LLM Inference

---

> > > ### Author Response · Authors · 2024-08-13
> > > **Comparison with Quest**
> > >
> > > Quest [1] shares similarities with ArkVale (and some other works like InfLLM [2]) in estimating and performing topk-filtering on kv-cache at the page/block granularity. The main differences are as follows:
> > > - Their approach aligns more closely with post-training dynamic sparse attention methods like SparQ [3] and IceFormer [4]. These methods do not involve eviction of kv-cache and cannot save GPU memory.
> > > - The estimation method we employ is based on bounding-volume, whereas their estimation method is similar to the "cuboid-max" (a subclass of our approach) introduced in our paper.
> > >
> > > [1] Quest: Query-Aware Sparsity for Efficient Long-Context LLM Inference
> > >
> > > [2] InfLLM: Training-Free Long-Context Extrapolation for LLMs with an Efficient Context Memory
> > >
> > > [3] SparQ Attention: Bandwidth-Efficient LLM Inference
> > >
> > > [4] IceFormer: Accelerated Inference with Long-Sequence Transformers on CPUs

---

### Author Rebuttal · Authors · 2024-08-06

## Adaption to other models

We adapt ArkVale to both `MaziyarPanahi/Llama-3-8B-Instruct-64k` (with 64k context-length extended from Meta Llama-3-8B) and `mistralai/Mistral-7B-Instruct-v0.3` (with a 32k context-length) in hugginggface, and test them on the datasets used in our paper, with the results shown in the table below (with page-size=32):

| Model            | \|  | Llama3     |           |        |           |           | \|  | Mistral    |           |           |        |       |
| ---------------- | --- | ---------- | --------- | ------ | --------- | --------- | --- | ---------- | --------- | --------- | ------ | ----- |
| *Cache Budget*   | \|  | ***Full*** | *4096*    | *2048* | *1024*    | *512*     | \|  | ***Full*** | *4096*    | *2048*    | *1024* | *512* |
| HotpotQA         | \|  | **42.94**  | **40.9**  | 38.01  | 37.39     | 32.54     | \|  | **49.37**  | 48.72     | **49.92** | 49.14  | 48.91 |
| NarrativeQA      | \|  | **16.77**  | 17.2      | 18.03  | 17.8      | **18.14** | \|  | **28.74**  | **28.61** | 25.87     | 25.34  | 23.59 |
| Qasper           | \|  | **11.0**   | **10.6**  | 10.03  | 9.5       | 8.71      | \|  | **41.58**  | **42.2**  | 41.44     | 39.24  | 36.9  |
| GovReport        | \|  | **28.91**  | **27.34** | 25.62  | 24.38     | 21.96     | \|  | **34.91**  | **32.84** | 31.59     | 29.69  | 25.85 |
| TriviaQA         | \|  | **89.91**  | 89.91     | 89.91  | **90.12** | 89.35     | \|  | **88.59**  | **88.94** | 88.94     | 89.38  | 88.7  |
| PassageRetrieval | \|  | **52.75**  | 59.75     | 61.25  | 62.0      | **64.91** | \|  | **98.0**   | **95.0**  | 94.5      | 90.5   | 82.0  |

From the data in the table, we can observe that in most cases, ArkVale can approach or even surpass the accuracy of the original model when the cache budget does not exceed 4096.

---
## Additional Baselines

We add comparisons with two open-sourced concurrent works, SnapKV [1] and InfLLM [2]. SnapKV, similar to works like H2O and TOVA, evicts KV-cache based on historical token information, but it only performs eviction during the prefill phase, thereby reducing overhead in the decode phase. InfLLM is a context expansion work that employs block-level memory units, which share similarities with ArkVale's page-level eviction/recall method. However, it generalizes memory units using several representative tokens (defaulting to 4) and, like streaming-llm, it retains a fixed number of initial-tokens and recent-tokens, employing the same positional encoding for tokens outside the recent-tokens.

We test these two baselines on LongBench based on the experimental setup and evaluation method detailed in Section 6.3 of our paper. The experimental results are presented in *Figure 1 of the attached PDF*. SnapKV nearly outperforms all other baselines, yet it is still not better than ArkVale overall. We attribute this anomaly to our dynamic selection of important tokens. InfLLM, similar to ArkVale, inspects and recalls some important tokens, and additionally preserves initial-tokens and recent-tokens, thus performing comparably or even better than ArkVale on datasets such as Qasper, GovReport, and TriviaQA. However, since it apply the same positional encoding to all the tokens other than recent-tokens, its performance is unstable, as evidenced by its poor performance on NarrativeQA and PassageRetrieval. Furthermore, using representative tokens as page digest may not be good enough for page importance estimation.

*Figure 2 of the attached PDF* shows the average latency per decode step for ArkVale (page-size=32) and baselines under different sequence lengths in the passkey retrieval task. Due to the inconvenience of setting batch-size greater than 1 for some baselines, experiments were conducted with batch-size=1. In all cases, ArkVale achieves the shortest latency, which gradually increases with the increasing cache budget. H2O, TOVA, and StreamingLLM incur token eviction during every decode step, introducing additional overhead, whereas SnapKV only performs eviction during the prefill phase, resulting in better performance than the others. Although importance estimation and page recall may introduce some additional overhead to ArkVale, ArkVale is slightly faster than SnapKV, mainly due to its efficient paged memory management with page-attention. The poor performance of InfLLM is likely due to its suboptimal implementation of memory management and hand-crafted attention kernel.

[1] Infllm: Unveiling the intrinsic capacity of llms for understanding extremely long sequences with training-free memory

[2] Snapkv: Llm knows what you are looking for before generation

---
## Impact of Different Page-sizes

Generally speaking, the smaller the page-size, the more accurate the estimation of page importance, the higher the model's accuracy. However, a smaller page-size will also increase the number of pages for the same number of tokens, thereby increasing the space occupied by the page digest (1~2 token per page), and simultaneously increasing the latency overhead for page importance estimation.

We test the impact of different cache-budgets and page-sizes on model accuracy and decode latency on LongChat-7b-v1.5-32k. We select PassageRetrieval as the benchmark. The results are presented in *Figure 3 of the attached PDF*. It can be observed that as the page-size decreases/cache-budget increases, both accuracy and latency gradually increase. Notably, the accuracy for page-size=8 is slightly worse than that for page-size=16, possibly due to the smaller page-size causing fewer adjacent tokens to be selected simultaneously, which impacts attention locality. Considering accuracy and latency & memory overhead, we typically set the page-size to 16 or 32.

---

### Decision · Program_Chairs · 2024-09-25

**Decision:**

Accept (poster)

**Comment:**

This paper looks into the problem of KV cache compression for more memory-efficient LLM inference. Existing eviction-based KV cache compression methods permanently evict tokens which may later turn out to be important as generation moves forward. The paper introduces a method to offload a subset of tokens that are not important for the time being but can recall those tokens when needed if token importance changes dynamically. The paper also takes into account of page attention, which supports easy swap-in and swap-out of tokens to second memory during inference. Overall, the reviewers are positive to the paper. The main reasons for accepting the paper are:
1) It seems to be novel to add the mechanism of dynamically retrieving tokens back from secondary memory based on the change of token importance.
2) Promising empirical speedups.
3) Relatively comprehensive evaluation.

There were some concerns from reviewers about the generalizability of the approach to other models and comparison with additional baselines. However, those were addressed via the rebuttal. The paper can be improved if it can incorporate the discussion of concurrent work, as suggested by reviewer YdSG.